

# A wind proxy based on migrating dunes at the Baltic Coast: statistical analysis of the link between wind conditions and sand movement.

Svenja E. Bierstedt[1], Birgit Hünicke[1], Eduardo Zorita[1], and Juliane Ludwig[2]

[1]Institute for Coastal Research, Helmholtz-Zentrum Geesthacht, Geesthacht, Germany
[2]Institute of Geology, University Hamburg, Hamburg, Germany

*Correspondence to:* Svenja Bierstedt (svenja.bierstedt@hzg.de)

**Abstract.** We statistically analyse the relationship between the structure of migrating dunes in the Southern Baltic and the driving wind conditions over the past 26 years, with the long-term aim of using migrating dunes as proxy for past wind conditions at interannual resolution. Dunes as wind proxies are not a totally new idea to the scientific community, but existing studies have so far analysed the link of dune structure and wind only on temporal resolutions of decades or millennia. The present analysis

is based on the dune record derived from geo-radar measurements by Ludwig et al. (2016). The dune system is located at the Baltic Sea coast of Poland and is migrating from west to east along the coast. Ludwig et al. (2016) suggested that the analysed dunes show an alternation in the sediment composition that can be used to determine the annual migration velocity which can be seen as a wind proxy. Here, we present a detailed statistical analysis of this record and calibrate it as a wind proxy. To our knowledge there are no adequate, homogeneous meteorological station data for this area available to validate this proxy.

Therefore we based our analysis on a gridded regional meteorological reanalysis data set (coastDat2) over the recent decades. We include precipitation and temperature into our analysis, in addition to wind, to learn more about the dependency between these three atmospheric factors and their common influence on the dune system. We set up a statistical linear model based on the correlation between the number of days with west and south-west wind directions above a pre-defined wind speed threshold and the dune migration velocities. To some extent, the dune intervals can be seen analogous to a tree ring widths,

and hence we used a proxy-validation method usually applied in dendrochronology when the available meteorological record is short, namely the cross-validation with the leave-one-out-method. This revealed correlations between the wind record from the reanalysis and the reconstructed wind record derived from the dune structure in the range of 0.28 and 0.63. Thus, our study verifies that this type of dunes can be validated with dendrochronological methods and derive acceptable validation values as a wind proxy.

The identified link between the dune annual layers and wind conditions from the meteorological reanalysis was additionally supported by the co-variability between dune layers and sea-level variations in the Southern Baltic Sea. Baltic Sea level variability in winter time is known to be strongly driven by westerly winds over this region. These results, therefore, provide an independent support, solely based on observations, of the link between annual dune layers and prevailing wind conditions.



# 1 Introduction

Future climate change may induce changes in wind conditions at all time scales, ranging from multi-decadal trends to changes in the daily and seasonal variability including wind extremes. To analyse future wind changes it is essential to understand how changes in the external forcing influenced wind conditions in the past. In contrast to other meteorological parameters like tem-

5 perature or precipitation, there is a dearth of wind proxy records capable of reflecting changes in past wind regimes. Any new proxies, albeit imperfect, can be very useful in this regard. Recently, Ludwig et al. (2016) presented a new proxy for annual wind-field variations based on a composite bar code, of a dune system alongshore the Polish Baltic Sea close to Łeba reflecting the width of different dune intervals that are annually formed. This present study statistically analyzes the link between this new proxy record, and wind conditions in order to evaluate and calibrate the potential to reconstruct past wind conditions.

Aeolian processes are the result of interactions between climatic factors like temperature, precipitation or wind and land surface. Therefore it can be assumed that coastal dunes include information about changes of these atmospheric parameters (Lancaster, 1994). The analysed dunes migrate and are composed of alternating layers with varying sediment and grain-size properties, which contain information about how the dune structure responded to past wind conditions during migration. These varying sediment properties can be seen as an analogous to tree-ring-width records, which also may include information about

changing climate conditions (Girardi, 2005). This kind of dunes do not only exist at the Polish coast, but also e.g. at the Curonian spit (Lithuania) where the alternating dune structure was also interpreted as a result of winnowing of lighter quartz grains due to higher wind speeds (see Sect. 2.2.2) (Buynevich et al., 2007).

The potential of dunes to provide information about storminess has already been demonstrated (Clemmensen et al., 2014; Costas, 2013; Reimann et al., 2011), but existing studies have used the connection between dune structure and wind only on

decadal or millennial temporal resolution. The novelty of the Ludwig et al. (2016) study lies in the focus on seasonal to annual resolution.

Proxy-based reconstructions of wind conditions may offer the advantage of a better temporal homogeneity over long-periods compared to observational records, for which changes in the location of the measure device may result in very large step-changes in the mean wind and wind variability. In addition, dune-based records may help fill spatial gaps in observational data

sets, which might be of special interest for analyses of a changing wind climate. For example, many wind analyses using pressure measurements as wind proxy, to take advantage of the more homogeneous properties of pressure readings over time (e.g. Alexandersson et al., 2000; Krueger et al., 2013). Other studies try to infer information of past wind events from documented damages on dikes (De Kraker, 1999) or forests (Nilsson et al., 2004).

Many studies have addressed past changes in wind climate (see review by Feser et al., 2015, and references therein) based on

different approaches. Some analyse wind speed changes (among others Alexandersson et al., 2000; Gulev et al., 2001; Wang et al., 2006; Matulla et al., 2007; Krueger et al., 2013) and some wind direction changes (e.g. Jaagus and Kull, 2011; Lehmann et al., 2011), both on different time scales and with different data sets. These studies include analyses of observations derived from instrumental records (Franzén, 1991; Chaverot et al., 2008), which are likely not totally consistent due to instrumental changes or relocations.





Other studies have used meteorological reanalysis, which should in principle overcome the inhomogeneity problem, instead of direct station observations (e.g. Gulev et al., 2001; Wang et al., 2006). Meteorological reanalysis is a data product constructed with a combination of weather information (e.g. surface weather stations, satellites, etc.) of past weather records, and simulations with a meteorological forecast model. These simulations assimilate the available observational records to produce a

5 gridded, homogeneous model data set of many atmospheric and oceanic variables with a temporal resolution of a few hours (Dee et al., 2015). Due to their use of observations, these kind of data may span a limited period in which the records can be considered homogeneous. However, the connection to observations is an advantage, as, in theory, meteorological reanalysis being close to the available - possibly sparse and incomplete - observation records, provide a multivariable data set that is complete in space and time.

Here, we study the statistical links at interannual time scales between wind conditions in different seasons and the annual dune intervals, and assess the relationships between the reconstructed wind and wind characteristics with a focus on wind direction and wind speed. We include precipitation and temperature into our analysis to learn more about the dependencies of these three atmospheric factors and their influence on the dune system. Although the analysed dune archive covers only a short period from 1987 to 2012 we consider this analysis relevant for the paleoclimate community as a proof-of concept to derive wind

proxies once longer dune records with annual resolution become available. Hence, this analysis could be adopted to other dune systems which are bigger and or move more slowly, e.g. at the Curonian spit.

This paper is structured as follows: Chapter 2 describes the analysed reanalysis data and the Łeba dunes. Chapter 3 explains the used statistical methods. Chapter 4 presents the results. A discussion of the results and a conclusion closes the manuscript.

## 2 Data and area

In the following the reanalysis product coastDat2, used in this study, is introduced and briefly discussed. Furthermore, the investigation area Łeba and its climatological and dune characteristics are described. An elaborated description about the analysed dune data can be found in Ludwig et al. (2016).

### 2.1 Meteorological data

For the main investigation, wind data from the regional meteorological reanalysis data set coastDat2 (Geyer, 2014) is used. The

25 meteorological reanalysis is the result of a model simulation in which observations have been assimilated. In contrast to the operational reanalysis used for real weather forecasts, in which the model has been continuously improved through time, the meteorological model is the same for the whole period of reanalysis. The number of observations assimilated in the reanalysis may change over time as well, but since one of the aims of meteorological reanalysis is to produce a homogeneous data set, the changes in the number and coverage of observations is usually kept to a minimum, at least over several decades. For reanalysis

that span longer periods, like the 20CR product covering the last 150 years, this condition cannot be fulfilled. In our case, the coastDat2 data set covers the period from 1948 onwards. Thus, it can be considered to be largely homogeneous.

In this study, the analysed data set is coastDat2, a result of a regional climate simulation with the non-hydrostatic operational



weather prediction model COSMO-CLM in CLimate Mode (COSMO-CLM, Rockel and Hense (2008)) driven by meteoro-
logical initial and boundary conditions from the global low-resolution NCEP/NCAR Reanalysis 1 data (1948-present; T62
$(1.875°, \approx 210 km)$, 28 level, (Kalnay et al., 1996; Kistler et al., 2001)). The regional simulation covers Europe and was con-
ducted, applying spectral nudging (after von Storch et al., 2000). It has a spatial resolution of $0.22°$ and the output is available

on hourly temporal resolution. However, the information derived from the dunes cannot provide such high temporal resolu-
tions, hence, we decided that daily averaged data is sufficient.

Weidemann (2014) compared the measured wind conditions at three German coastal stations (Kiel, Warnemünde, Kap Arkona)
with the COSMO-CLM (used to generate coastDat2) model output. He reported a slight systematic overestimation of wind
speed, but good agreement regarding daily mean wind speeds, generally acceptable results regarding the daily wind speed

variability - notwithstanding an underestimation of high wind speeds - and comparable wind directions. He stated that some
discrepancies might be introduced by the model COSMO-CLM, but that other differences between model and observations
might be due to in-situ measurement errors.

Although meteorological reanalysis are close to observations, it can be argued that they are still a model product. Unfortu-
nately, the joint analysis of the dune layers and observed winds is hampered by the lack of direct nearby observations in this

area. To verify our reanalysis-based results with real observations, we resort to other observations that are known to be related
to seasonal wind conditions in the Baltic Sea, and compared observed coastal sea level data from various stations across the
Baltic Sea with dune layer thickness. Baltic Sea level variations are strongly driven by surface winds (Hünicke et al., 2015),
especially in autumn and winter, and hence can be seen as a good proxy regarding wind in this region. The reason for not using
direct observational data for the calibration and verification of our analysis are the mentioned inhomogeneities of such data

sets especially with regard to wind information. And although there exist a weather station close to the analysed dunes this
station is located in the woods, which compromises especially north-western wind information (Ludwig et al. 2016).

The relationship between wind and sand migration may additionally be dependent on other atmospheric parameters, like pre-
cipitation and temperature. Regarding temperature, we used the daily mean 2-meter temperature from coastDat2. Regarding
precipitation we used the daily sum of total precipitation from coastDat2, which includes convective and large-scale precipi-

tation as well as snow. The results obtained with coastDat2 data were also confirmed with precipitation data provided by the
Climate Research Unit (CRU; Mitchell and Jones, 2005). This latter gridded data set is the result of a spatial interpolation of
station data. The obtained results were found to be comparable (not shown).

## 2.2   Łeba dunes

The active dune system in Łeba (Poland) covers an area of 5,5 $km^2$ and is situated on top of a barrier that separates the Lake

Lebsko from the Baltic Sea. To the north, pine trees and foredunes prevent sediment supply from the beach to reach the proper
dune system; hence, the material that forms the dune is self-contained with little contamination from outside the system. Public
entering is prohibited since 1967.

The barchanoid dunes are up to 600 m long and 27 m high. The sands are fine-grained and well-sorted and the dunes attain
an average migration velocity of around 10 m/yr. This dune system has been analysed before by Borówka (1979, 1980, 1995);





Borówka and Rotnicki (1995). These authors also mentioned some climatological characteristics of this area, which will be briefly recapped and compared to our own results in the following subsection. Additionally, an overview about their results and results from Ludwig et al. (2016) regarding the relation of wind and this dune structure will follow.

### 2.2.1 Climatological characteristics

Due to its west-east alignment, the dune migration is strongly connected to westerly winds (Borówka, 1980), which mostly occur and are strongest during winter and autumn. Westerly winds transport the sand from the luv side (west) of the dune to the lee side (east) of the dune and so contribute to the eastward movement of the dune (Fig. 1) due to sediment deposition on its lee side. Hence, the stronger or more frequent westerly winds are, the more sand is transported to the lee side of the dunes, which also leads to a higher dune migration velocity.

Additionally to wind, temperature and precipitation have an influence on the Łeba dune migration, e.g. frost and precipitation might stabilise the dune and hinder the sand transport. Nevertheless, winds are the most important drivers of aeolian processes in general. Borówka (1980) reported the mean annual total precipitation amount in the research area to be about 700 mm with a maximum occurring in summer and autumn. Furthermore, he stated that the area undergoes only small annual temperature variations. From coastDat2, within the period 1948-2012, we calculated a mean annual precipitation amount of about 630 mm

and seasonal mean temperatures for winter=$-1.6$°C, spring=$5.4$°C, summer=$15.7$°C, autumn=$7.9$°C averaged over the area shown in the right panel of Figure 2.

### 2.2.2 Coastal dunes as archive of seasonal wind intensity

The dune sands are characterised by alternating changes in the sediment composition. The dune structure shows intervals dominated by light quartz grains and intervals with interspersed intervals of heavy minerals. Both intervals are caused by seasonally

changing wind conditions (Borówka, 1980). The quartz interval consists predominately of quartz grains with dispersed heavy minerals. Quartz grains as well as heavy minerals are mobilized along the luv-side of the dune and transported to the east by westerly winds, which are stronger and occur more frequently during autumn and winter. In contrast, winds from the east winnow quartz grains, as unveiled by, Borowka's observations of the Łeba dunes, leaving enriched heavy minerals behind. This gives rise to an alternating structure of layers that can be investigated with the help of ground-penetrating radar (GPR). GPR

has already been used to analyse dunes by (e.g. Bristow and Goodal, 1996; Clemmensen et al., 1996).

Ludwig et al. (2016) showed that a quartz-dominated interval and an interval enriched in heavy minerals represent a whole year. This alternating pattern is termed sedimentary bar code (Fig. 1), and its thickness varies from year to year. The link between layer thickness and wind is not-linear, as the grain mobilization requires winds to surpass a certain threshold. Also, the effect of the winds on a particular dune may depend on rather local and individual characteristics of the dune. As in the case of

30 other proxy records, and in order to overcome dune-to-dune variations and gaps in the sedimentary record Ludwig et al. (2016) analysed a dune complex composed of six dunes, providing individual bar codes that were later complied into one composite bar code for the entire dune-field, and applying two dendrochronological methods, namely replication and cross-dating. Annual variations in the bar code thickness, and hence in the migration rates, correlate with changing west wind intensities. A




comparison between observational wind data, from a station located near the sample side, and the bar code thickness showed that during years with strong west winds the net dune progradation to the east is higher than during years with weaker west wind intensities. The bar code record covers a time period of 26 years from 1987 to 2012.

In this study, we provide a more detailed statistical validation of the link between dune structure and wind conditions by in-

5 vestigating the seasons and wind directions for which the dune structure can be considered more representative of the wind conditions, estimate optimal wind thresholds from the data and provide an amount of wind variance that can be derived from the dune records and provide uncertainty ranges if the dune records were used to reconstruct past wind variability.

## 3 Statistical methods

This study focuses mainly on the relationship between sand movement and wind conditions during different seasons: Winter

(December to February; DJF), Spring (March to May; MAM), Summer (June to August; JJA) and Autumn (September to November; SON). The analysed wind conditions are defined based on wind speed thresholds. The thresholds relevant for sand movement at the investigation side are unknown, so that different thresholds have been considered as a free parameter to find an optimal relationship between wind conditions and the dune bar code. We also use eight different wind direction subdivisions (North=N, North-East=NE, East=E, South-East=SE, South=S, South-West=SW, West=W, North-West=NW), 360 degrees are

divided into eight equal sectors of 45 degrees each to derive conclusions on the dune driving wind directions.

We set up a linear regression model in which the independent variable is the migration velocity of white, black and both combined intervals derived from the layer thickness and the dependent variable is the number of days with daily wind means from a certain direction and above a predefined wind speed threshold. In this way we identified the leading relationships between the white and black bars (predictor–y) and different combinations of wind direction and wind speed (predictand–ỹ).

This model is tested and validated with the help of cross-validation, namely the leave-one-out-method (Michaelsen, 1987; Birks, 1995). This validation technique is commonly used for dendrochronological analysis to investigate the linear relation between tree ring width and temperature when the temperature record is short.

The leave-one-out-method addresses the problem of a short record of observations that does not leave enough independent data for validation, once all data have been used to calibrate the statistical model. In the leave-one-out method to validate the

25 statistical model, all observations except one are used to calibrate the statistical parameters. The calibrated statistical model is then used to estimate the value of the predictand for the left-out observation, which is then compared to that observation. A complete loop over all observations is then conducted in which at each step only one observation is not included in the calibration of the statistical model. In the end, a measure of the statistical skill is obtained as an average of the mismatch between estimated and observed values of the predictand at each 'left-out' time step. In our case, this means that successively one of the

30 available 26 predictor values (bar thickness) is left out and the remaining 25 values are used to "predict" the corresponding days per wind direction over a pre-defined wind speed threshold (predictand). In the end we have got 26 predicted wind condition values (ỹ), which can be compared to the actual values (act(y)) derived from coastDat2. We compare both with the help of the root-mean-square-error (rmse–Eq. 2), which can be used to determine the explained variance ($rmse^2$) and we calculate the





correlation coefficient between predictand and actual values.

In addition, we try to find an optimal ratio between the number of westerly and easterly winds that better describe the thickness of the black interval. Thereby, we use a local regression (loess regression) - where local means here that the regression is based only on a set of observational data points (x,y) - that lie within a certain limited region in a x-y plot. The value of the parameters

of the statistical model thus depend on the value of predictor and predictand x,y. The statistical models that we have used in this analysis are weighted linear least squares and a 2nd degree polynomial model. This local regression is equivalent to finding a local and second-degree polynomial that better fit the (x-y) data points. The width of the loess window is optimized with the lowest root-mean-square-error after the leave-one-out-method.

$$\tilde{y}_i = p1_i * y_i + p2_i \qquad (1)$$

$$rmse = \sqrt{\frac{\sum\limits_{i=1}^{n} (act_i(y) - \tilde{y}_i)^2}{n}} \qquad (2)$$

## 4    Results

Before we applied the linear regression model to identify a relationship between dune migration and wind conditions, we

analysed the connection between the dune movement and other atmospheric parameters. The following section is devoted to the correlation between the migration velocities of the white and black intervals and temperature, precipitation and wind. Later, we explain the results concerning the linear regression model between the migration velocity and wind for a specified direction and speed threshold.

### 4.1    Dune migration velocity and meteorological forcing

As already mentioned, the dunes in the study area consist of alternating intervals with different sedimentary characteristics which are termed "bar-code". Quartz-dominated intervals are imaged by white bars and have a mean thickness of 6.37 m. The black bars (intervals) are characterised by heavy minerals and show an averaged thickness of 6.15 m. The dune migrates by the action of the wind as material from the luv-side of the dune is transported over the dune all the way forward to the lee-side of the dune. The succession of white and black intervals corresponds to the annual cycle in the meteorological characteristics

and this allows for the dating of each pair of intervals. In the study area, the time period covered by the intervals formed in the dunes span the years 1987-2012.

The thickness of both type of intervals varies, but not independently of each other, as the thickness of the black and white intervals correlates with 0.63. The whole dune system migrates 12.52 m per year on average. This dune migration is influenced by atmospheric parameters. These parameters are temperature, precipitation and wind. The most important parameter is obviously



the wind as it transports the sand. Nevertheless the other factors may have some influence. We investigated the relationship between bar thickness and seasonal precipitation. The amount of soil wetness influences the compactness of the top intervals of the dune and their sensitivity to the wind drag. The colder seasons winter (DJF) and spring (MAM) show slight positive correlations for both intervals, which indicates an increasing bar thickness during wetter periods. The temperature also shows

relations to the dune interval thickness. Autumn is the only season showing some non-negligible correlation for black intervals (0.33). Hence, in autumn, sand movement has a slight tendency to be faster with higher temperatures. The other seasons reveal no correlation between temperature and bar thickness.

The two meteorological parameters temperature and precipitation combined might play a role, especially during winter season. It is assumed that low temperatures, below zero, together with precipitation stabilise the dunes and thus hinder the sand trans-

10 port. To consider this effect, we analysed the correlation between wind conditions (number of days per wind direction) and sand movement by excluding or including days with frost and precipitation. The biggest differences can be seen in winter (compare Fig. 4a and 3a), some differences in spring (compare Fig. 4b and 3b) and none in summer and autumn (not shown). Winter and spring show with and without frost days the same correlation sign, but some correlations are higher for days without frost and precipitation. In spring higher correlations can be seen for northern and eastern winds, but changes are still quite small.

For winter, the correlation coefficients get lower (higher) without frost and precipitation, especially for white (black) bars and E (SE) winds. Nevertheless, autumn still has the highest correlations between wind and bar thickness.

#### Wind

The analysis regarding the relationship between wind conditions and intervals is based on wind intensity per wind direction.

The latter is divided into eight subdivisions (N, NE, E, SE, S, SW, W, NW). The wind condition is defined by applying two measures. 1. The measure is the mean wind speed per wind direction calculated only in the days with mean wind from that particular wind direction. 2. The measure is the number of days per wind direction. The correlations between white, black and combined bar thicknesses and these two wind condition measures for the eight predefined wind directions are shown in Fig. 5 and Fig. 4, respectively. The correlation coefficients reveal summer to be the least effective season for sand transport

regardless of the wind condition definition. For the other seasons there are some differences depending on the definition of wind conditions used: In spring we only see noticeable correlations for E winds for the black interval using the mean wind speed definition (r≈0.3). The mean wind speed from E apparently has an influence on the thickness of the black interval. The number of days from a particular wind direction seems to be less effective in spring. This is an interesting result as it is an indication for the winnowing of white and black grains as mentioned by Borówka (1979). In winter, the link between dune

layers and wind clearly depends on the definition of wind condition. The layer thickness is positively correlated with mean wind speed for almost all wind directions and intervals.However, the correlation between layer thickness and the number of days from particular directions displays opposite correlation, with eastern and northern wind directions showing negative and western and southern wind directions showing positive correlations. These opposite correlations can be also seen for autumn, especially for the black interval. Hence, in autumn and winter the strong winds prevent the winnowing effect described in the

introduction. In these seasons the wind speed of easterly winds seems to be high enough to not only erode the lighter white



material but also the black heavy minerals. Autumn is the season with highest correlations for both measures and both bars, pointing at this season as the most important for sand transport.

As a next step, we analyse days per wind direction with wind speeds over a predefined wind speed threshold to connect the two measures based on wind speed and based on days per wind direction. The wind speed is binned into 10 groups ranging from

0 to >10 m/s. The wind directions with non-negligible correlation coefficients are E and NE during spring (Fig. 6c+d), W and SE during winter (Fig. 6a+b) and W, SW, NE during autumn (Fig. 7). The correlation coefficients in summer and in the other wind directions are predominantly low (not shown).

In spring we see a difference in the sign of the correlation coefficients between dune layers and E and NE winds. For E winds the correlation is positive especially for the white interval. NE winds show negative correlations for all layers. Winter also

shows contradicting signs in the correlations to SE and W winds, although with smaller correlation differences between white and black layer.

Concerning the variations in the wind threshold, both winter and spring show higher correlations for wind speeds above 4 m/s. In autumn the correlations are highest for a threshold of 8m/s of NE winds for the black interval, with a negative sign, and for a threshold between 3m/s and 5m/s of SW winds and for a threshold of 5 m/s of W winds, both with a positive sign.

## 15    4.2    Linear regression

The highest correlation between wind and the thickness of the white and black interval can be seen in autumn for SW wind direction, thus we use this season and direction to set up a linear regression model with the interval thickness as predictor and wind speed as predictand. For SW winds correlations are highest for wind speeds from 3 to 5 m/s (s. Fig. 7b). The linear relationship between days per wind direction and within this wind speed band and the migration velocity of black and white

interval is tested with the leave-one-out-method (Sec. 3). We use the migration velocity (predictor), and its linear relation to the number of days with SW wind with the above mentioned wind speed (predictand). The leave-one-out-method allows for the validation of this relation by comparing the predictand with the actual number of days per wind direction. Table 1 shows the correlation coefficients between the predicted and actual values, the root-mean-square-error and the explained variance of this analysis are shown.

The best results are obtained for SW winds, which is likely due to the wind speed threshold being more strictly defined. With these threshold values the correlation between migration velocity and number of days per wind direction are higher (compare e.g. Fig. 7a,b). However, one has to keep in mind that a higher wind speed threshold (3-5 m/s) excludes many observations. This validation of the regression model to predict SW winds from the dune layers shows comparable results to accepted validation values of dendrochronological analyses.

We see a strong positive (negative) connection between the migration velocity of the black interval and the number of days with W/SW (E/NE) winds. We assume that white and black sands are transported together eastwards by westerly winds and this explains the positive correlation between the black intervals and the number of days with W and SW winds. In days with easterly winds, which are usually weaker than westerly winds, only the white lighter particles are transported to the back of





the dune, enriching the black interval. This explains the negative correlation between the number of days with E and NE winds and the black layer.

This idea of winnowing was already explained by Borówka (1979) for the Łeba dunes.

As already mentioned, westerlies transport white and black particles together to the east, where they deposit and build a new

interval. Easterly winds on the other hand winnow only the lighter white grains and transport them backwards to the west, hence a black interval forms. This effect suggests that there might be an optimal difference of days with west and east winds per year that results into a thicker black interval.

To test this hypothesis we use a scatter plot between the difference in the number of days with west (W, SW, NW) and east (E, SE, NE) winds during all seasons and the black interval thickness per year (s. Fig. 8). The data points are smoothed with

a loess filter (red line in Fig. 8). If an optimal ratio between east and west should exist, the smoothed curve would show a clear maximum. Our result does not show this maximum and therefore no clear optimal ratio can be derived from these results. Nevertheless, there seems to be a minimum value of this ratio ($\approx 4500$) under which the black interval tends to be small.

### 4.3 Baltic Sea level and dune layers

Our results are based on reanalysis data. These are, as described before, derived due to a combination of a numerical model

and observations. The reason for not using station data for the calibration and verification of our analysis are the mentioned inhomogeneities of such data sets especially with regard to wind information.

However, to show the correlation between the dune layer thickness and wind, we used - as an indirect wind measure - Baltic Sea level data. Coastal Sea level in the Baltic Sea in autumn and winter is strongly driven by the intensity of the westerly winds, so that there exists a strong correlation between seasonal mean sea-level and many indices of westerly wind intensity, such as

the North Atlantic Oscillation (e.g. Andersson, 2002). Figure 9 shows the correlation pattern between the total thickness of the dune layer and mean winter sea-level in the tide-gauges provided by the Permanent Service for Mean Sea Level (PSMSL). Since we are investigating the correlations at interannual time scales and sea-level records, which are affected by the long-term sea-level rise and long-term crust movement, the sea-level time series have been linearly detrended prior to the calculation of the correlations. Positive correlations between autumn (SON) Baltic Sea level and dune layer thickness are found for most

tide-gauges. The sign of this correlation is also consistent with the idea that strong westerly winds generally cause thicker dune layers. In addition, the correlation patterns display higher correlation at tide-gauges that are located closer to the dune site and lower correlations for tide-gauges located further apart. The only plausible explanation that may explain this spatial pattern of correlations between coastal sea-level variations and the thickness of dune layers is a link between both through the common wind forcing, Therefore, this strengthens our results derived from the reanalysis data set coastDat2.

### 30 5   Discussion and conclusion

Our analysis validated migrating coastal dunes identified by a geological analysis of Ludwig et al. (2016) as a new wind proxy at interannual time scales. To validate this wind proxy against wind observations we chose to use a reanalysis data set, in-



stead of data from a meteorological station. There are two main reasons for this choice. One is that the main application of our study is the reconstruction of wind conditions that may have a wider spatial representativity than station data. Climate reconstructions will eventually be compared with climate model simulations, which have a spatial resolution comparable to meteorological reanalysis, and therefore it is more convenient for this purpose to assure the statistical link between the proxy

record and regional, as opposed to pointwise, wind data. This allows for a larger scale reconstruction than the observational data from one station, used by Ludwig et al. (2016) to identify the wind proxy. A second reason is that we consider that data from only one station can be affected by inhomogeneities in time, and also by local conditions like the presence of forest and hence maybe not representative of the wind conditions forming the dunes, e.g. Ludwig et al. (2016) also state that some wind directions seem to be under-represented in the station data due to the station position located behind trees.

Nevertheless, in order to demonstrate the credibility of our results compared to observations, we additionally calculated correlations between observed Baltic Sea level data and dune layer thickness during autumn (SON). Hünicke et al. (2008) stated that wind is the main driver of interannual Baltic Sea level variations, hence there should be a direct link between both parameters. Therefore, Baltic Sea level can be seen as a good proxy for wind variability in the Baltic Sea region. The derived correlation values showed a clear positive link between dune layer thickness and Baltic Sea level across the whole Baltic regions and thus

a clear positive link to wind as well. This result confirms the relationship between wind and dunes identified in our study with the help of the reanalysis coastDat2.

The analysed dunes at the Polish Baltic Sea coast are characterised by alternating white and black bars representing light quartz grains and heavy minerals. These bars might be regarded as analogous to tree ring width records. The analysis of the dune records composite was conducted similarly as with dendrochronological methods. Hence, the chosen validation technique

is also a common tool to verify the relation between tree-ring width and temperature.

We investigated the relationship between bar thickness and the atmospheric parameters: precipitation, temperature and wind. The focus lies on the relation to wind conditions because wind is assumed to actually transport the sand grains. However, precipitation and temperature also have an influence on dune migration:

Regarding precipitation, the results showed positive signs for the white and black bars for winter (DJF) and spring (MAM).

Borówka (1980) stated that some rain might improve the transport due to turbulence, which makes more sand grains available. We argue that the influence of precipitation on sand transport, and hence the dune processes, depends on the seasonal wind conditions. Ludwig et al. (2016) describe a secondary dune on top of the primary dune consisting of the white and black interval. This secondary dunes seem to be affected by precipitation due to erosion. This idea is supported by our results and shows that in wetter seasons the secondary dunes might be eroded into the primary dune and hence results in thicker dune intervals.

Due to its west-east alignment the dune is most sensitive to westerly (W, SW) and easterly (E, NE) winds. This relationship with wind depends on season and on direction. During winter and autumn, westerly winds correlate positively with dune interval thickness, whereas the easterly winds correlate negatively, more or less independently from wind speed. In spring there are positive correlations for eastern wind direction for the white interval.

After analysing the influence of meteorological parameters on dune migration, we focused on the linear relationship of the

migration velocity and the frequency of days with SW winds surpassing a specific wind speed threshold. The derived linear



relationships were validated with the leave-one-out method due to the limited length of the observational record. This linear model allowed to hindcast the wind speed from the migration of the dunes over the past decades. The correlations between the observed and reconstructed wind speeds lie between 0.28 and 0.63 and are comparable to the correlations typically obtained for other climatic proxies e.g. tree rings. As an example, Bräuning and Mantwill (2004) derived correlation values with leave-

5  one-out validation of 0.41 to 0.78. This results lead us to the conclusion that alternating dune structures can be used as wind proxies also on annual time scales.

Dunes as wind proxies had already been used before (Clemmensen et al., 2014; Reimann et al., 2011), but only on decadal or millennial temporal resolution. Our study therefore, statistically validates the interpretation by  of the dune intervals and dune migration velocities as indicators of annual and even seasonal wind conditions. Although, the dune system analysed here

10  covers only a period of 26 years, we suggest that the analysis is of relevance for paleoclimate studies since it can be applied to other dune systems covering longer time periods.

*Acknowledgements.*  This work is a contribution to the Helmholtz Climate Initiative REKLIM (Regional Climate Change), a joint research project of the Helmholtz Association of German research centres (HGF). The authors would like to thank Sebastian Lindhorst for explaining dune mechanisms.



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

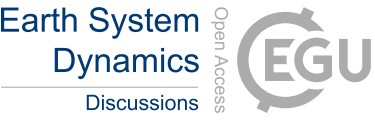

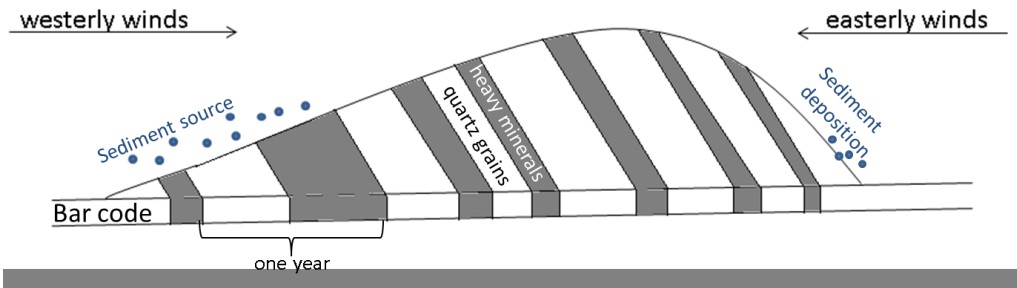

**Figure 1.** Schematic representation of of the Łeba dune structure (adapted from Ludwig et al. (2016)).





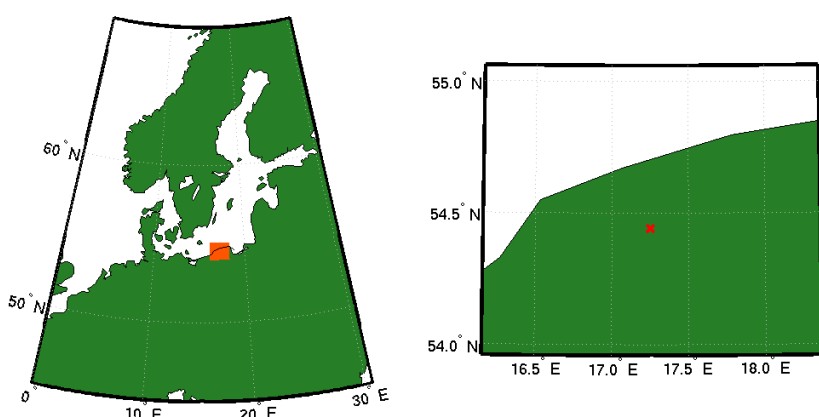

**Figure 2.** Location of investigation area. Left: Red box marks analysed gridded wind information from coastDat2 (1987-2012). Right: Analysed area with dune location (red dot) close to Łeba, Poland.





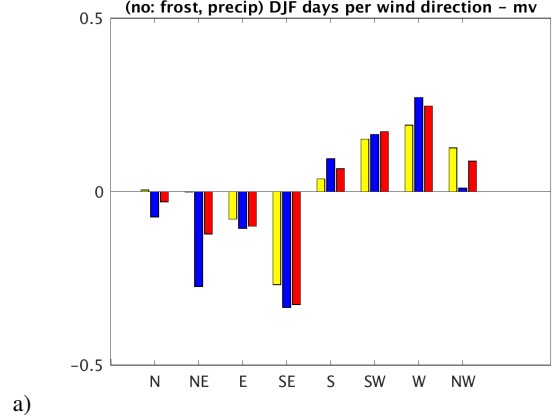
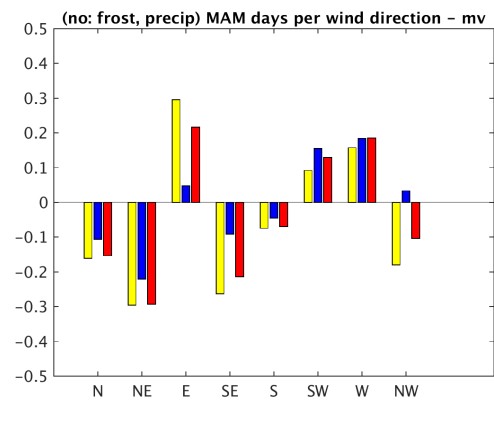

a)                                                                b)

**Figure 3.** Correlation between the number of days per wind direction without frost and precipitation days and dune thickness of the white interval (yellow), black interval (blue) and both together (red). The correlations are shown for the seasons winter (DJF), spring (MAM) and for eight wind directions.



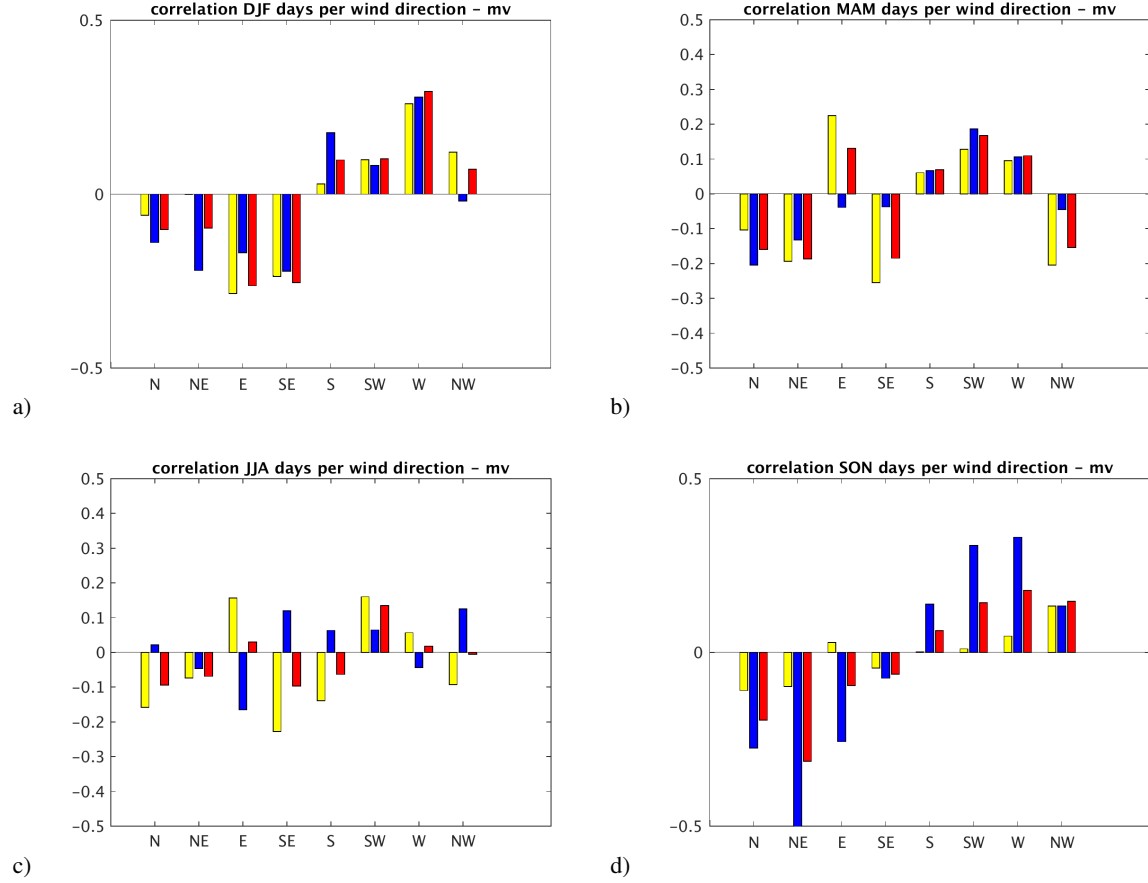

**Figure 4.** Correlation between the number of days per wind direction and dune thickness of the white interval (yellow), black interval (blue) and both together (red). The correlations are shown for the seasons winter (DJF), spring (MAM), summer (JJA) and autumn (SON) and for eight wind directions.





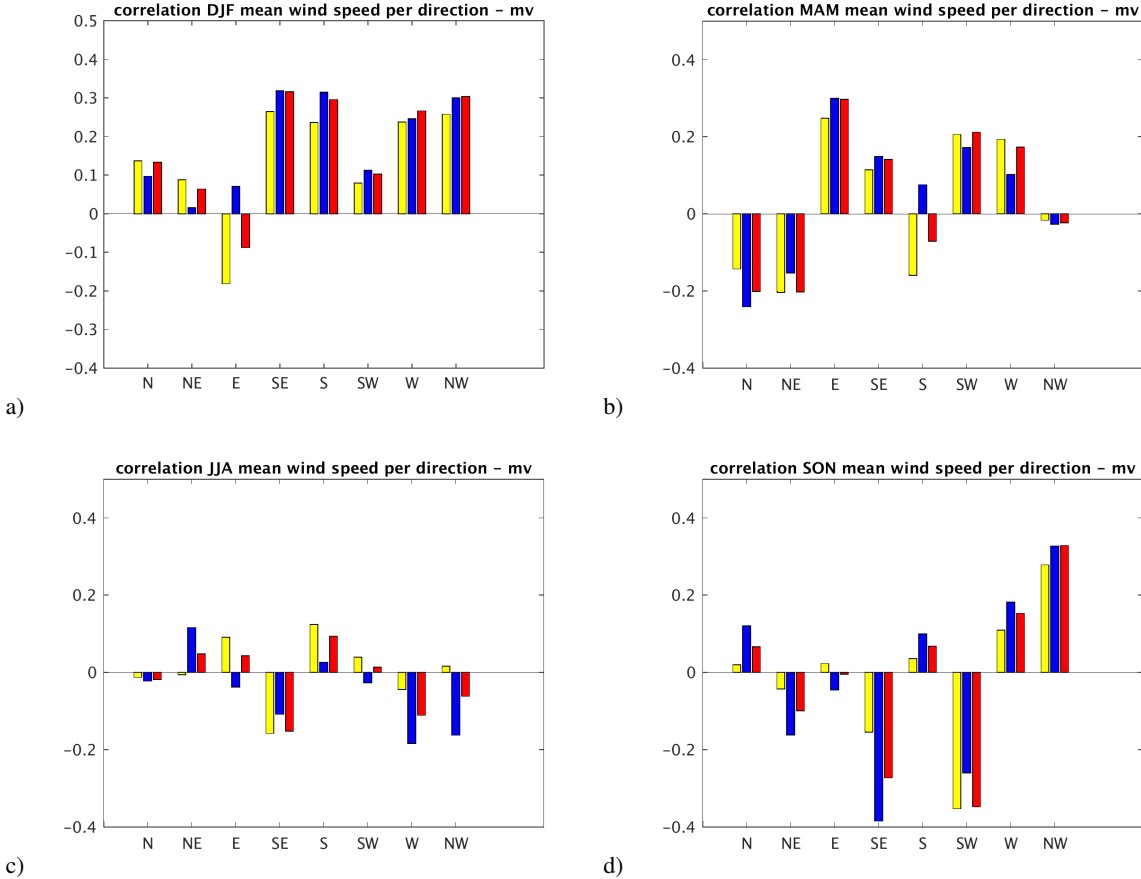

**Figure 5.** Correlation between mean wind speed and dune thickness of the white interval (yellow), black interval (blue) and both together (red). The correlations are shown for the seasons winter (DJF), spring (MAM), summer (JJA) and autumn (SON) and for eight wind directions.





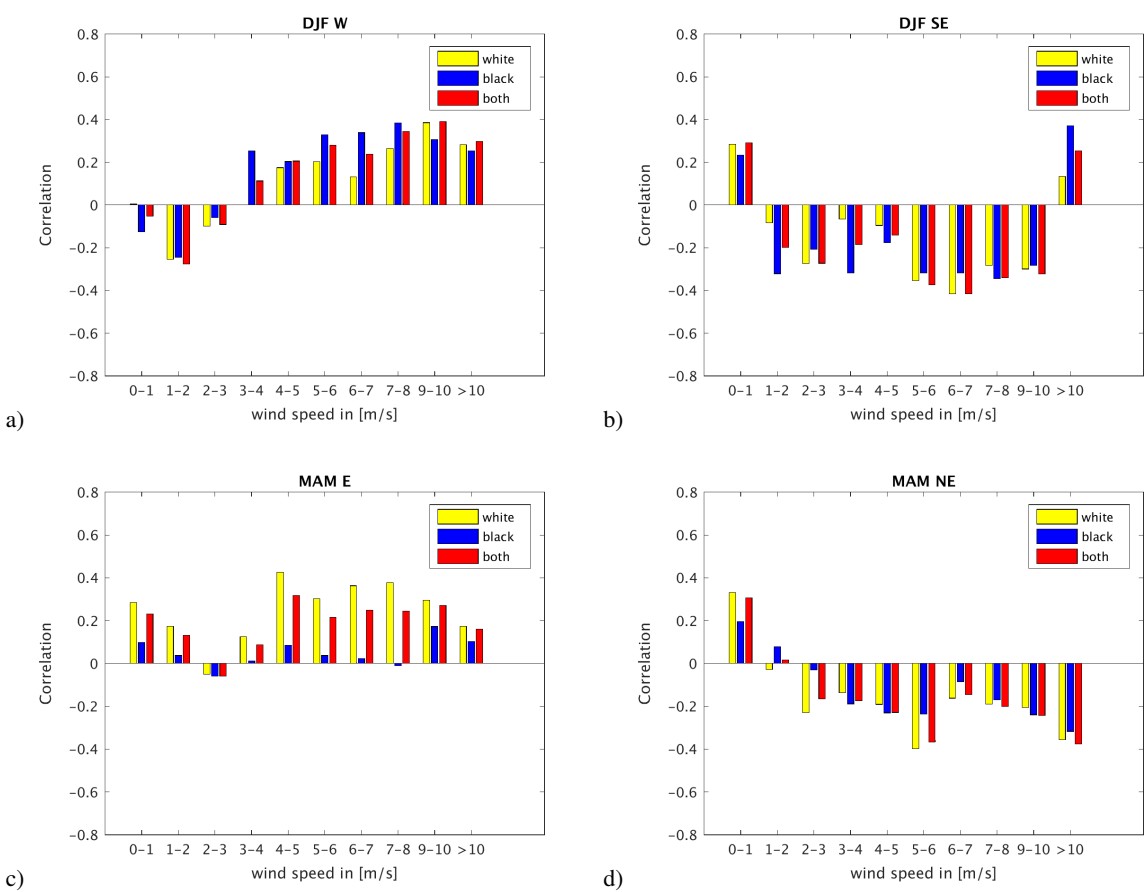

**Figure 6.** Correlation between the number of days per wind direction in a specified range of wind speeds and dune thickness of the white interval (yellow), black interval (blue) and both together (red). The correlations are shown for the seasons winter (DJF; a+b) and spring (MAM; c+d) for W (a), SE (b), E (c) and NE (d) wind directions.




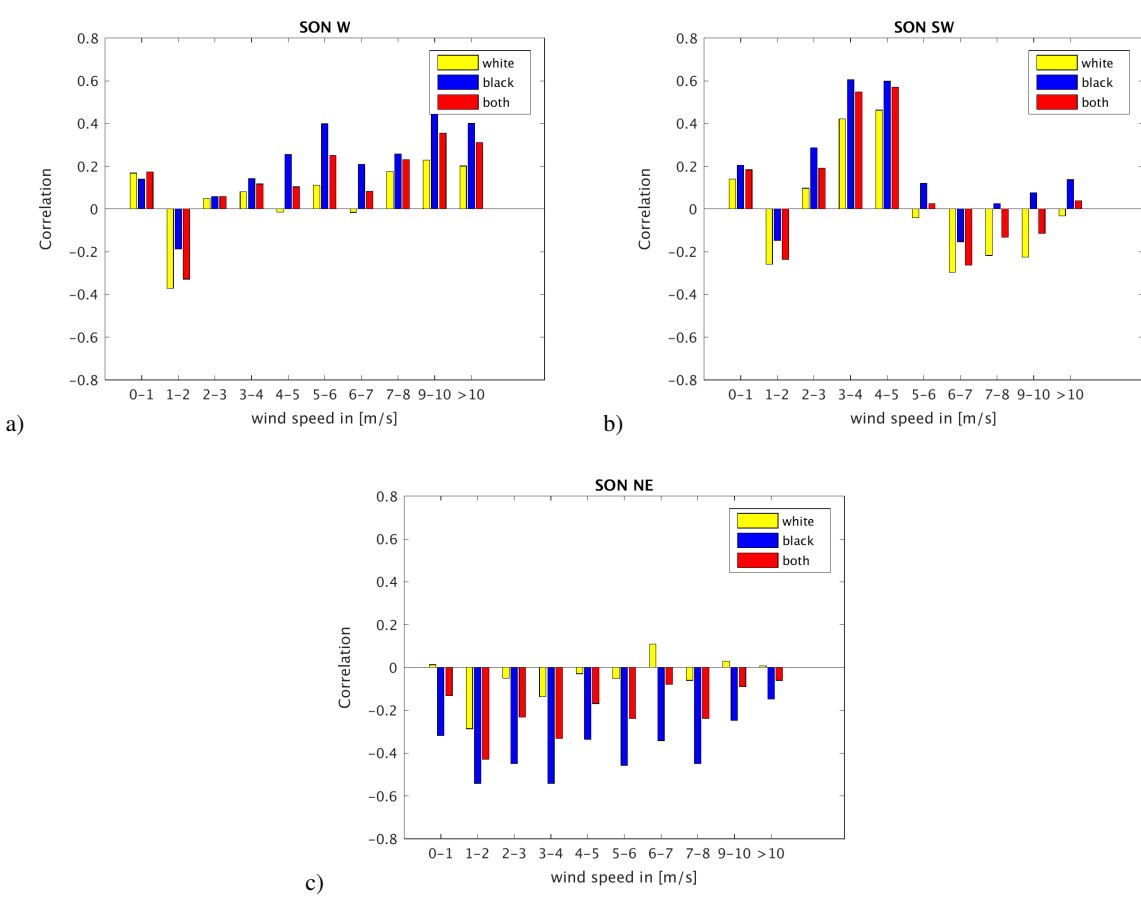

a)

b)

c)

**Figure 7.** Correlation between the number of days per wind direction in a specified range of wind speeds and dune thickness of the white interval (yellow), black interval (blue) and both together (red). The correlations are shown for the seasons autumn (SON) for W (a), SW (b) and NE (c) wind directions.



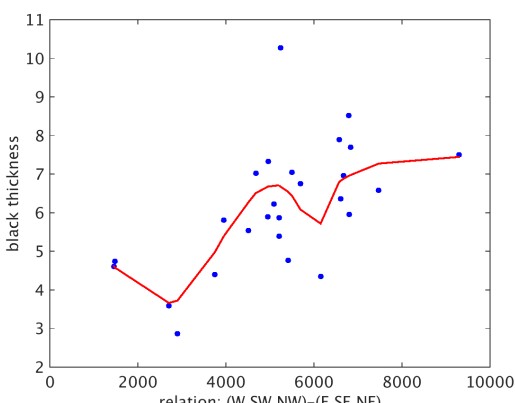

**Figure 8.** Scatter plot of the difference between westerly (W, SW, NW) and easterly (E, SE, NE) winds and the black interval thickness. The red line shows the smoothing with a loess filtering (s. Sec. 3).





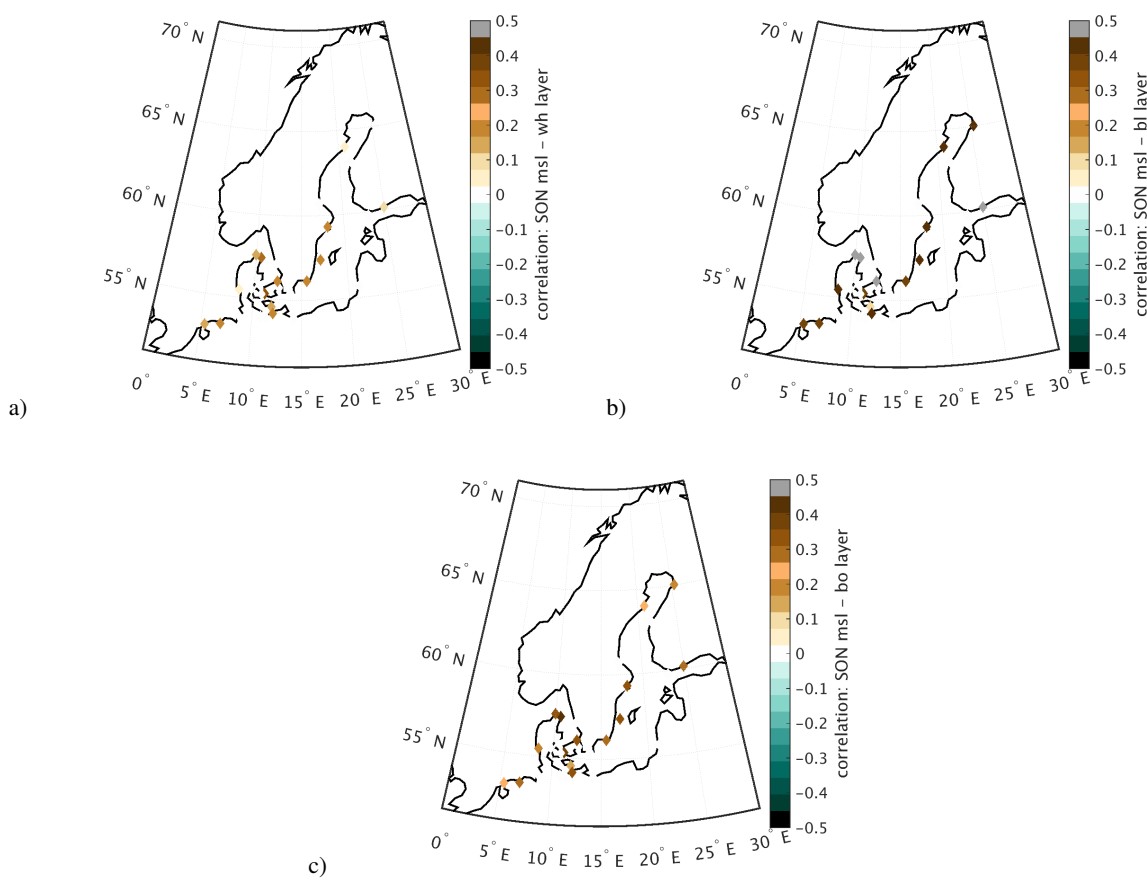

**Figure 9.** Correlation between mean sea level data from Baltic coastal stations and the dune layer thickness values during autumn (SON) for a) the white layer, b) the black layer and c) both layers combined.





**Table 1.** Correlation, root-mean-square-error and explained variance values (obtained with leave-one-out validation) used to compare predicted and actual number of days per wind direction. The prediction is based on dune thickness, which is identified to have a linear relation with SW winds between 3 and 5 m/s.

|  |  | correlation | rmse | exp. variance |
|---|---|---|---|---|
|  | white | 0.28 | 1.93 | 8.07% |
| SW | black | 0.63 | 1.56 | 39.21% |
|  | both | 0.52 | 1.70 | 27.16% |