# Peer review of "A wind proxy based on migrating dunes at the Baltic Coast: statistical analysis of the link between wind conditions and sand movement."

_Earth System Dynamics, 2016_

## Referee Comment (RC1) · T. Soomere (Referee) · 18 Jan 2017

General comments

This is, in principle, an interesting piece of work that addresses a new option of construction of a proxy of wind regime in the past using the formation of sand layers of active dunes with different properties under anisotropic wind climates. Even if the pattern of correlations of wind properties, Baltic Sea water levels as their proxy and features of sand layers remains quite noisy, the message itself is valuable and worth of further elaboration.

Unfortunately several flaws in the manuscript defocus the outcome and diminish the

implications of the analysis. It is still likely that the manuscript will become publishable after major revision.

Specific major aspects

1. No information is presented about statistical significance of the presented correlations. Even though this way of categorisation of links between different quantities is basically formal and may provide false positive or negative conjectures, it is important to amend the material so that the strength of correlations is characterised by some other kind of measure with clear interpretation. Also, in some occasions other measures such as root mean square deviation or similar would be helpful to clarify the context.

2. The formulation of the main outcome (the existence of positive/negative correlations) is interpreted in a manner that is not really supported by the presented analysis. This first of all applies to wording like "validation" of the results. Even though the correlations in question are interesting and potentially valuable, essentially no validation is provided in the manuscript.

3. It is questionable whether it is acceptable to rely so strongly on a source (Ludwig et al. 2016) that is currently not yet published and has thus not fully undergone quality control of the peer-review process. A partial solution would be to make this source available, e.g., via ArXiv.org or similar channels.

4. The use of English is generally fine but the use of several specialised terms is questionable and in some locations help from a native speaker might make the presentation more transparent and compact. The text contains numerous unnecessary repetitions, redundant words and phrases. The examples below highlight only a small selection of these, so the text should be thoroughly and entirely polished. The Abstract is far too long and contains several exact sentences from the body of the manuscript. The Discussion is basically a reformulation of work done, including assumptions, and contains no proper discussion. The list of references systematically ignores capitalisation

of proper names.

Technical issues and recommendations for Sections 1 and 2 (pages 1–5)

Page 1, sentences on lines 3–4, 6–8, and 9 should not be part of Abstract and the rest of Abstract should also be compactified.

Line 18: the expression "that this type of dunes can be validated with dendrochronological methods and derive acceptable validation values" requires complete rewriting.

Line 20: " from the meteorological reanalysis" is redundant.

Lines 21–24: the two sentences are redundant (or packed into a few words) without any loss to the content.

Page 2, line 2: remove "Future" as the statement is valid also for the past and present

Line 13 "during migration" is redundant.

Line 15: " This kind of dunes do not only exist at the Polish coast, but also ..." should be rephrased.

Line 20: it is questionable whether it is appropriate to appraise here a source that is not yet published.

Line 23 "measure device" should be corrected; also probably "step-like" is meant.

Page 3, lines 1–2: "instead of direct station observations " is redundant without any loss of content.

The sentence on lines 7–9 is weakly (if at all) connected with the text and could be deleted without any harm to the core message.

Line 10: "statistical link" sounds strange.

Line 11 and in many occasions below: the term "dune intervals" is, to my knowledge, not widely used in coastal science. Its classic (but perhaps partially outdated) notion

is the distance between crests of two subsequent dunes. Its use in the context of this manuscript may collide with the use of "interval" in a completely different sense as intervals between glacial cycles. Thus, I strongly recommend to consider another term, and if "dune intervals" is generally used in this field of science, to bring a thorough explanation and references in order to avoid misinterpretation.

Line 11: "the relationships between the reconstructed wind and wind characteristics" is incomprehensible.

Line 12: say just "speed" instead of "wind speed".

Lines 20–22: such short introductions to chapters are basically fine, but they also serve as partial repetitions of the material in Introduction and could be merged with the information on lines 17-18.

Line 25: the sentence is pure repetition of material on lines 3-4 and should be omitted.

Line 29: "changes" requires "are".

Lines 30–31: the length of the coastDat2 does not make it homogeneous, so please reformulate the claim.

Line 32: the text says already fourth time here that coastDat2 is used in the presented manuscript.

Page 4, line 1: "COSMO-CLM" should appear only once.

Line 3: a comma before the equality sign should be deleted, and one should say "levels".

Line 5 " on hourly temporal resolution": say simply "hourly".

Line 8: delete " (used to generate coastDat2)" as this information was already provided.

Line 10 and several other occasions: "comparable" is not a good adjective to characterise the match of two quantities; please use some more exact quantification.

The sentence on lines 18-20 seems not to carry any new information and could be deleted.

Line 20: consider replacing "And although there exist" by "Although there exists".

Line 22: consider replacing "be dependent" by "depends".

Lines 23-24: remove " Regarding temperature," and " Regarding precipitation".

Line 25: remove " obtained with coastDat2 data" as it is clear from the context from where the results stem.

Page 5, lines 5–6: consider replacing the unclear phrase " which mostly occur and are strongest" by "which are most frequent and strongest".

Lines 7–8: delete "due to sediment deposition on its lee side" as this is clear from the first part of the sentence.

Lines 11–12: the claim " Nevertheless, winds are the most important drivers of aeolian processes in general" is true by definition (Aeolus was the Greek god of the winds) and should be removed from here.

Line 12: remove "amount".

Line 15: please provide standard deviations of the seasonal and annual temperatures; otherwise the claim that the area undergoes only small annual temperature variations is not justified.

Lines 18–19: as above, please consider whether "interval" is the appropriate term here; also rephrase "intervals with interspersed intervals of heavy minerals".

Lines 21–22: the sentence almost exactly repeats material on lines 6-7.

Lines 24–25: remove " GPR has already been used to analyse dunes by ".

Line 27: it is not clear what "its thickness" means (of layers, not the code?).

Line 28: either "not linear"(without a hyphen) or "non-linear"; please notice that these categories are different.

Line 29: remove "rather".

Line 31: consider saying "cluster" instead of "dune complex composed".

Page 6: "progradation" generally cannot be "higher" but should be faster (or its rate may be higher).

Sections 3–5 the manuscript contain a more or less similar proportion of small issues per page but I do hope that they will all be removed in the substantially revised version of the paper.

---

## Referee Comment (RC2) · Anonymous Referee #2 · 16 Feb 2017

Review of Bierstedt et al. – A wind proxy based on migrating dunes at the Baltic Coast: statistical analysis of the link between wind conditions and sand movement

Our understanding of past decadal to long-term changes of wind statistics is very limited. Longer observations or robust wind reconstructions are mostly rare or absent. One idea to overcome this problem is to use Aeolian transport of dust or sand particles as a proxy for variations in atmospheric circulation and wind climate. Different often local applications such as grain size analysis in sediment cores from peat bogs or dunes etc. have been used. Making use of the layered structure of sand dunes, the authors test here statistically to which extent the annual layers can be linked to driving atmospheric variables, predominantly wind.

In my opinion, the manuscript offers an interesting evaluation of such a sand layer structure as a temporally high resolution proxy for wind. With a correlation of up to 0.63, the barcode structure of the sand dunes provides a valuable wind proxy. While the novelty of using such a barcode structure as wind proxy and reasonable correlations are promising, the paper requires some significant re-work wrt. to the clarity of text, the paper's structure and the robustness of statistical evaluation. I therefore think the manuscript may be published after major revisions.

Major comments:

Text: The structure of the manuscript, the motivation for this study and the scientific context should be improved (see below).

Statistics: The authors present an interesting evaluation but without a proper validation. Although the sample size is relatively small, significance testing (p-values) would be important to provide at least some estimate about the robustness of the results/methods. The author's attempt to test the link between temperature or precipitation and sand mobilization/transport and hence a potential disturbance of the wind signal by other factors is, not fully convincing. The causal link remains unclear here whether temperature and precipitation simply co-vary with wind or really affect sand mobilization. This is a serious problem for most other paleo-wind proxies as well as e.g. dry periods may be confused with stormy periods or wet periods simply co-vary with windy periods etc. This aspect should be addressed or at least discussed in more detail. The use of a LOESS regression is not convincing given the low sample size and unclear relation. Correlations for the whole wind season should be added as the barcode has an annual resolution.

Specific comments:

Abstract:

One or two leading sentences, why it is important to study the wind climate in the past

and why it is important to use proxies even for shorter timescales, would make the paper more appealing to read. The abstract should be improved by focusing on main aspects of your work only, without too much detail.

Introduction:

Generally, the structure here is a bit chaotic. I would recommend to re-write it following a clear structure like why it is important to study the wind climate, what has been done already using which data or methods (the references on page 2-3), what are the conclusions and open questions from these studies so far (see e.g. Rutgersson et al. 2015; Feser et al. 2015; Christensen et al. 2015 for reviews), the problems of the data/methods and need for using different proxies, how Aeolian transport can be used here followed by here we present. . . or similar.

Page 2, line 1-2: References needed, many studies investigated it already (e.g. Christensen et al. 2015 for references)

Page2, line 4: References needed, this has been done already (e.g. Rutgersson et al. 2015; Feser et al. 2015)

Page 2, line 23-24: Not only the location changes, a reference could be e.g. Lindenberg et al. 2012

Page 2, line 29 to Page 3, line 9: This is all quite technical for an introduction. What do these studies tell us about past wind climates? What are their main conclusions so far? What are open questions a dune proxy may help to answer?

Page 3, line 2-9: A detailed explanation for this data product is not really needed here. A short note might be more interesting here like e.g. that even the use of long-term reanalysis data like 20CR has been shown to be problematic regarding long-term trends (Krueger et al. 2013 etc.) and dune records may therefore help to get more consistent results.

Data and area

Acknowledging the geological context of a site-specific analysis, I would suggest a classical structure 2.1 Area/Leba Dunes, 2.2 Climatological characteristics, 2.3 Meteo data, 2.4 Dunes

2.1. Meteorological data

I think it is enough to write "we use a numerically downscaled reanalysis dataset…" with two or three sentences which reanalysis and regional model, spectral nudging and resolution used in coastdat2. For details you can refer to Geyer 2014. More relevant for this study are the properties and validation of simulated wind (Weidemann 2014).

Page 3: line 29: "usually kept to a minimum". This is not correct, all available data is used for each time step. Reanalysis hence gives the best possible estimate for each time step which may lead to artefacts if the type, quality or number of observations changes over time. Rather a frozen data assimilation scheme is used to minimize these effects. For your region and time period, all the issues are not really relevant after ∼1980.

Page 4: line 12: The imperfect NCEP forcing and its coarse resolution is another relevant source of error here

Page 4, line 33: Could you give the typical size here (dominant sand fraction, $\mu$m or a range)?

Page 5, line 16: What about wind? Reference here to Ludwig et al. for more detailed seasonal information (Fig. 5 in that paper)

Page 5: line 28: Is there not any experimental / theoretical range giving a rule of thumb which wind speed can mobilize which grain size or mass? If so, you could use it to physically verify the realism of your statistically estimated thresholds.

Page 5, line 31: compiled

Statistical Methods

It is convincing and very nicely shown, that one barcode interval reflects one year in Ludwig et al. However, the separation into junks of three months for wind is in the end subjective and artificial wrt. to the annual dune activity. Consider e.g. that your seasonal correlation analysis may suffer from intra-seasonal changes in wind activity over time (e.g. Lehmann et al. 2011). I would hence suggest to add first a correlation analysis for the whole wind year (e.g. ONDJFM). You could then replace the JJA figures in the multi plot figures 4 and 5 with the full wind season. Then you can continue and show also whether the full wind season or rather a fraction of the season yields the highest correlation.

The same applies for the wind directions. The correlations for the wind octants in combination with the small sample size in this study may be quite sensitive to small random changes to the neighboring octant (as can be anticipated from the wind rose figures in Ludwig et al.). I think you should test in addition quadrants of $90°$ (e.g. W= 225-315° or SW=180-270°). It is certainly interesting to make the detailed tests in this study but one aim should be to find the optimal setting with the best fit to the dune data rather than limiting it to very strict seasons and directions.

Page 7: line 2: Please explain in more detail how such a ratio or difference might look like and why, I cannot follow here. What are the x-axis units in Fig. 8?

Page 7, line 3: I see the point of exploring the outcome of a LOESS fit here. But the result does not look useful. Based on Fig. 8, a linear regression (digitizing your data in Fig. 8, I got y=0.00053x+3.388; $r^2$=34%; p=0.0012) looks more convincing although it remains unclear to me, what it means. With the low sample size, LOESS regression is very sensitive to outliers. As it does not yield any equation, the fit cannot be reproduced by others without having the original data. I would therefore suggest to stick to a linear fit, give confidence intervals and explain the outcome.

Page 8, line 3: "slight positive correlation" – Which value? Give a p-value.

Page 8, line 4: "indicates an increasing bar thickness during wetter periods". And how

does that match with soil wetness and compactness mentioned before? I think this only tells you that more storms co-vary with more rain, but there is no causation more rain = more sand transport. This should be at least discussed if the low sample size does not allow a comparison like drier storm seasons vs. wetter storms seasons in comparison to the barcode. Maybe you could make a quick test for your period if/how wind above your chosen threshold is correlated with precipitation and temperature.

Page 8, line 5: "non-negligible" – Please give a value and p-value here. Why only the black?

Page 9, line 17: "this season and direction" – It makes sense to use the best combination but you should reconsider whether the best combination might not be the full wind season (e.g. SONDJFM) as mentioned before. As you mentioned the dunes in Lithuania, it would make sense to also provide your regression model for re-use or reproduction of results.

Page 10, line 2: To which extent could you use the deviation of the black-white ratio from being relatively equal ($\sim$1) to say sth. about years of more easterly or more westerly years? I did not really get that point from the manuscript.

Page 10, line 10-13: This fit makes little sense. Please replace LOESS with a linear fit and give the equation, $r^2$ and p-value (should be very close to what I wrote above). There is indeed no optimum (why should there?) but a linear fit is highly significant. What could that mean?

Page 10, line 17-20: The link between wind climate and sea-level is a bit more complicated depending on the region and timescale of wind/sea-level co-variations. The description here is too vague and some references should be given in addition. Note that most readers do not know anything about sea-level variations of the Baltic Sea.

Page 10, line 29: And how does the link of the wind forcing look like= How can it be explained?

Page 11, line 15-16: Rather than being suspicious about coastdat2 here, I would high-light that the positive link to sea-level is very useful as sea-level data goes further back in time than reanalysis and might be also more reliable than spurious trends in 20CR (Krueger et al. 2013), which do not affect yet the short period in this study.

Page 11, line 25-26: Very speculative. With "some rain" it might be true but not with more rain. If the "some rain" effect would be important, you should expect to get a negative correlation in your evaluation, but it is positive. I would add that more rain might just co-vary with more windy conditions. The mentioned erosion is also a very good point here.

Page 12, line 5: The p-values and adding an analysis of the full wind season ONDJFM might lead to an even more robust conclusion.

Figures:

Fig. 3+4+5: Use consistent tick marks on the y-axis. What means "- mv" in the figure titles? For all bar charts, you could consider using white, black and grey for white, black and mixed intervals. This would make it more intuitive.

Figure 8: I would rather use a linear fit. What are the units on both axis?

Figure 9: If possible, use bigger symbols for the gauge locations.

Table 1: Why not give the regression model (slope, intersect) in addition, also p-values?

Additional references:

Christensen, O. B.; Kjellström, E. & Zorita, E.: Projected Change—Atmosphere. In: The BACC II Author Team (Eds.): Second Assessment of Climate Change for the Baltic Sea Basin, Springer International Publishing, 2015, 217-233, doi:10.1007/978-3-319-16006-1_11

Lehmann, A.; Getzlaff, K. & Harlaß, J. (2011): Detailed assessment of climate vari-ability in the Baltic Sea area for the period 1958 to 2009. Clim. Res., 46, 185-196,

doi:10.3354/cr00876

Lindenberg J, Mengelkamp H-T, Rosenhagen G (2012): Representativity of near surface wind measurements from coastal stations at the German Bight. Met Z 21:99-106.

Rutgersson, A., Jaagus, J., Schenk, F., Stendel, M., Bärring, L., Briede, A., Claremar, B., Hanssen-Bauer, I., Holopainen, J., Moberg, A., Nordli, O., Rimkus, E., and Wibig, J.: Recent Change – Atmosphere. In: The BACC II Author Team (Eds.): Second Assessment of Climate Change for the Baltic Sea Basin. Springer International Publishing, 2015, 69-97, doi:10.1007/978-3-319-16006-1_4
* * *

---

## Author Response (AR1)

We would like to thank the reviewer Tarmo Soomere for his carefully reading and the very constructive criticism. We respond to all comments below.

**RC1 (T. Soomere):**

Specific major aspects

1. No information is presented about statistical significance of the presented correlations. Even though this way of categorisation of links between different quantities is basically formal and may provide false positive or negative conjectures, it is important to amend the material so that the strength of correlations is characterised by some other kind of measure with clear interpretation. Also, in some occasions other measures such as root mean square deviation or similar would be helpful to clarify the context.

> We included new figures where a '*' marks significant correlations.
> We use RMSE to compare the results derived from the linear regression model and the actual values from coastDat2.

2. The formulation of the main outcome (the existence of positive/negative correlations) is interpreted in a manner that is not really supported by the presented analysis. This first of all applies to wording like "validation" of the results. Even though the correlations in question are interesting and potentially valuable, essentially no validation is provided in the manuscript.

> We do not totally understand this comment. We use the term validation in the usual statistical meaning, i.e. the comparison between predicted and observed values in an independent data set. Our validation measures are included in Table 1 where the results of the linear regression model are compared with actual values from coastDat2. We acknowledge that the length of the record is short, and this is why we cannot split the data set in a calibration and independent validation period. This is the reason why we used the leave-one-out method. This comparison is to our knowledge an accepted (see tree ring record validations) validation measure. Perhaps the reviewer refers to another meaning of the term validation, including for instance a mechanistic explanation or comparison with a dynamical model. We now explain in the text more clearly our usage of the term validation.

3. It is questionable whether it is acceptable to rely so strongly on a source (Ludwig et al. 2016) that is currently not yet published and has thus not fully undergone quality control of the peer-review process. A partial solution would be to make this source available, e.g., via ArXiv.org or similar channels.

> Unfortunately, this is currently not possible. The manuscript is still under revision in Aeolian Research. This unfortunately precludes the publication in another platform, but we now refer to the PhD dissertation by Juliane Ludwig that is published and available on-line at the repository of the University of Hamburg

4. The use of English is generally fine but the use of several specialised terms is questionable and in some locations help from a native speaker might make the presentation more transparent and compact. The text contains numerous unnecessary repetitions, redundant words and phrases. The examples below highlight only a small selection of these, so the text should be thoroughly and entirely polished. The Abstract is far too long and contains several exact sentences from the body of the manuscript. The Discussion is basically a reformulation of work done, including assumptions, and contains no proper

discussion. The list of references systematically ignores capitalisation of proper names.

> We carefully re-read the text and hope to make the necessary adjustments to avoid redundant repetitions. The abstract was rephrased and shortened. We reworked the discussion and changed capitalizations in the references.

Technical issues and recommendations for Sections 1 and 2 (pages 1–5)

Page 1, sentences on lines 3–4, 6–8, and 9 should not be part of Abstract and the rest of Abstract should also be compactified.

> We rephrased and shortened the abstract.

Line 18: the expression "that this type of dunes can be validated with dendrochronological methods and derive acceptable validation values" requires complete rewriting.

> Old version: "Thus, our study verifies that this type of dunes can be validated with dendrochronological methods and derive acceptable validation values as a wind proxy."

> New version: "The revealed correlations between the wind record from the reanalysis and the wind record derived from the dune structure is in the range between 0.28 and 0.63 yielding similar statistical validation skill, as dendroclimatological records. "

Line 20: " from the meteorological reanalysis" is redundant.

> We changed this accordingly.

Lines 21–24: the two sentences are redundant (or packed into a few words) without any loss to the content.

> We reworked the abstract in large parts and these sentences are no longer included.

Page 2, line 2: remove "Future" as the statement is valid also for the past and present

> We changed this accordingly.

Line 13 "during migration" is redundant.

> We removed "during migration".

Line 15: " This kind of dunes do not only exist at the Polish coast, but also ..." should be rephrased.

> Old version: "This kind of dunes do not only exist at the Polish coast, but also e.g. ..."

> New version: "Comparable dune systems can also be found at other coasts e.g. ..."

Line 20: it is questionable whether it is appropriate to appraise here a source that is not yet published.

> Old version: "The novelty of the Ludwig et al. (2016) study lies in the focus on seasonal to

annual resolution.”

New version: “The reconstruction by Ludwig (2017) is the first attempt to use dunes as wind proxies on seasonal to annual resolution. ”

Line 23 "measure device" should be corrected; also probably "step-like" is meant.

Old version: “...changes in the location of the measure device may result in very large stepchanges in the mean wind and wind variability.”

New version: “...changes in the location of the measuring device may result in very large abrupt artificial changes in the mean wind and wind variability (Krueger, 2014). ”

Page 3, lines 1–2: "instead of direct station observations " is redundant without any loss of content.

We deleted "instead of direct station observations"

The sentence on lines 7–9 is weakly (if at all) connected with the text and could be deleted without any harm to the core message.

Maybe we did not phrase this sentence clearly enough. This sentence is meant to be connected to the previous sentence that demonstrates the  disadvantage of the reanalysis. The following sentence should, on the other hand, present the clear advantage of reanalysis data (homogeneity) compared to station data. We tried to rephrase it more clearly:

Old version: “Due to their use of observations, these kind of data may span a limited period in which the records can be considered homogeneous. However, the connection to observations is an advantage, as, in theory, meteorological reanalysis being close to the available - possibly sparse and incomplete - observation records, provide a multivariable data set that is complete in space and time.

New version: “Due to their use of observations, the time span covered by reanalysis is also limited. On the other hand, the connection to observations may be advantageous because meteorological reanalysis data aim to track real observational data, in contrast to free-running model simulations that do not include data assimilation. ”

Line 10: "statistical link" sounds strange.

We changed “links” to “relationship”.

Line 11 and in many occasions below: the term "dune intervals" is, to my knowledge, not widely used in coastal science. Its classic (but perhaps partially outdated) notion is the distance between crests of two subsequent dunes. Its use in the context of this manuscript may collide with the use of "interval" in a completely different sense as intervals between glacial cycles. Thus, I strongly recommend to consider another term, and if "dune intervals" is generally used in this field of science, to bring a thorough explanation and references in order to avoid misinterpretation.

We changed “dune interval” to “dune layer”.

Line 11: "the relationships between the reconstructed wind and wind characteristics" is incomprehensible.

Old version: "...the relationships between the reconstructed wind and wind characteristics..."

New version: "...the relationships between the reconstructed and actual wind characteristics, derived from the reanalysis,..."

Line 12: say just "speed" instead of "wind speed".

We changed this accordingly.

Lines 20–22: such short introductions to chapters are basically fine, but they also serve as partial repetitions of the material in Introduction and could be merged with the information on lines 17-18.

We agree with the reviewer. This might be merged into the last part of the introduction. Nevertheless, we personally think that a short "introduction" to the second section may be helpful to lead the readers, especially as this section includes several subsections with important information.

Line 25: the sentence is pure repetition of material on lines 3-4 and should be omitted.

We removed this sentence.

Line 29: "changes" requires "are".

We changed this accordingly.

Lines 30–31: the length of the coastDat2 does not make it homogeneous, so please reformulate the claim.

Old Version: "In our case, the coastDat2 data set covers the period from 1948 onwards. Thus, it can be considered to be largely homogeneous."

New Version: "The coastDat2 data set covers the period from 1948 onwards, and thus spans a period with an almost stable number of observations. Hence, it can be considered to be largely homogeneous."

Line 32: the text says already fourth time here that coastDat2 is used in the presented manuscript.

Old Version: "In this study, the analysed data set is coastDat2, a result of..."

New Version: "CoastDat2 is a result of..."

Page 4, line 1: "COSMO-CLM" should appear only once.

We changed this accordingly.

Line 3: a comma before the equality sign should be deleted, and one should say "levels".

We changed this accordingly.

Line 5 " on hourly temporal resolution": say simply "hourly".

We changed this accordingly.

Line 8: delete " (used to generate coastDat2)" as this information was already provided.

We changed this accordingly.

Line 10 and several other occasions: "comparable" is not a good adjective to characterise the match of two quantities; please use some more exact quantification.

We changed "comparable" in the mentioned occasion to "equal" and in the other cases to "similar".

The sentence on lines 18-20 seems not to carry any new information and could be deleted.

We deleted the mentioned sentence.

Line 20: consider replacing "And although there exist" by "Although there exists".

We changed this accordingly.

Line 22: consider replacing "be dependent" by "depends".

This is just personal taste. We would prefer to keep "be dependent".

Lines 23-24: remove " Regarding temperature," and " Regarding precipitation".

We changed this accordingly.

Line 25: remove " obtained with coastDat2 data" as it is clear from the context from where the results stem.

We changed this accordingly.

Page 5, lines 5–6: consider replacing the unclear phrase " which mostly occur and are strongest" by "which are most frequent and strongest".

We changed this accordingly.

Lines 7–8: delete "due to sediment deposition on its lee side" as this is clear from the first part of the sentence.

We deleted "due to sediment deposition on its lee side".

Lines 11–12: the claim " Nevertheless, winds are the most important drivers of aeolian processes in

general" is true by definition (Aeolus was the Greek god of the winds) and should be removed from here.

The sentence was removed.

Line 12: remove "amount".

We changed this accordingly.

Line 15: please provide standard deviations of the seasonal and annual temperatures; otherwise the claim that the area undergoes only small annual temperature variations is not justified.

We included standard deviations.

Lines 18–19: as above, please consider whether "interval" is the appropriate term here; also rephrase "intervals with interspersed intervals of heavy minerals".

All "intervals" were replaced.
Old version: "...intervals with interspersed intervals of heavy minerals "

New version:"...layers with interspersed deposits of heavy minerals."

Lines 21–22: the sentence almost exactly repeats material on lines 6-7.

Yes, it is a little repetition about the sand movement direction. However, this sentence also adds two new pieces of information. One is that both sand materials are transported together. The second is that  it provides the main  wind directions of transport and the seasons in which  they occur. Therefore we would like to keep the sentence as it is.

Lines 24–25: remove " GPR has already been used to analyse dunes by ".

We changed this accordingly.

Line 27: it is not clear what "its thickness" means (of layers, not the code?).

Old version: "This alternating pattern is termed sedimentary bar code (Fig. 1), and its thickness varies from year to year."

New version: "This alternating pattern is termed sedimentary bar code (Fig. 1), and the thickness of the individual bars varies from year to year."

Line 28: either "not linear"(without a hyphen) or "non-linear"; please notice that these categories are different.

We changed this to "not linear".

Line 29: remove "rather".

We removed "rather".

Line 31: consider saying "cluster" instead of "dune complex composed".

> We changed this accordingly.

Page 6: "progradation" generally cannot be "higher" but should be faster (or its rate may be higher).

> We changed "higher" to "faster".

Sections 3–5 the manuscript contain a more or less similar proportion of small issues per page but I do hope that they will all be removed in the substantially revised version of the paper.

> We re-read and reworked the manuscript and hope to have eliminated those small issues.

We would like to thank the reviewer and the editor for their carefully reading and the very constructive criticism. We respond to all comments below.

**RC2:**

In my opinion, the manuscript offers an interesting evaluation of such a sand layer structure as a temporally high resolution proxy for wind. With a correlation of up to 0.63, the barcode structure of the sand dunes provides a valuable wind proxy. While the novelty of using such a barcode structure as wind proxy and reasonable correlations are promising, the paper requires some significant re-work wrt. to the clarity of text, the paper's structure and the robustness of statistical evaluation. I therefore think the manuscript may be published after major revisions.

**Major comments:**

Text: The structure of the manuscript, the motivation for this study and the scientific context should be improved (see below).

Statistics: The authors present an interesting evaluation but without a proper validation. Although the sample size is relatively small, significance testing (p-values) would be important to provide at least some estimate about the robustness of the results/methods. The author's attempt to test the link between temperature or precipitation and sand mobilization/transport and hence a potential disturbance of the wind signal by other factors is, not fully convincing. The causal link remains unclear here whether temperature and precipitation simply co-vary with wind or really affect sand mobilization. This is a serious problem for most other paleo-wind proxies as well as e.g. dry periods may be confused with stormy periods or wet periods simply co-vary with windy periods etc. This aspect should be addressed or at least discussed in more detail. The use of a LOESS regression is not convincing given the low sample size and unclear relation. Correlations for the whole wind season should be added as the barcode has an annual resolution.

> Response to the general comments, further elaborated in the response to the specific comments:
> We include now the statistical significance of the correlations
> We did try to disentangle the role of temperature and precipitation from that of the wind by estimating correlations excluding or including frost days. The differences in those correlations were discussed as well.
> There seems to be a misunderstanding about the rationale of using the loess regression and we hope it is now more clear in the revised version

**Specific comments:**

Abstract:
One or two leading sentences, why it is important to study the wind climate in the past and why it is important to use proxies even for shorter timescales, would make the paper more appealing to read. The abstract should be improved by focusing on main aspects of your work only, without too much detail.

> We rephrased and shortened the abstract and hope it now reads more clearly and focused.

Introduction:
Generally, the structure here is a bit chaotic. I would recommend to re-write it following a clear structure like why it is important to study the wind climate, what has been done already using which data or methods (the references on page 2-3), what are the conclusions and open questions from these studies so far (see e.g. Rutgersson et al. 2015; Feser et al. 2015; Christensen et al. 2015 for reviews), the problems of the data/methods and need for using different proxies, how Aeolian transport can be used here followed by here we present...or similar.

>    We have re-structured the introduction according to the suggestion of the
>    reviewer.

Page 2, line 1-2: References needed, many studies investigated it already (e.g. Christensen et al. 2015 for references)

>    We have added a reference.

Page2, line 4: References needed, this has been done already (e.g. Rutgersson et al. 2015; Feser et al. 2015)

>    We have added a reference.

Page 2, line 23-24: Not only the location changes, a reference could be e.g. Lindenberg et al. 2012

>    We have added a reference.

Page 2, line 29 to Page 3, line 9: This is all quite technical for an introduction. What do these studies tell us about past wind climates? What are their main conclusions so far? What are open questions a dune proxy may help to answer?

>    We hope to have addressed these questions in the revised version.

Page 3, line 2-9: A detailed explanation for this data product is not really needed here. A short note might be more interesting here like e.g. that even the use of long-term reanalysis data like 20CR has been shown to be problematic regarding long-term trends (Krueger et al. 2013 etc.) and dune records may therefore help to get more consistent results.

>    Since this is a quite interdisciplinary work we think it is useful to explain
>    reanalysis data, even as early as in the introduction. The readers not used to
>    handle model data may miss the difference between free-running climate
>    simulations and meteorological reanalysis, and the links between the latter and
>    observations. Hence, we would prefer to keep this explanation.
>    However, we added the mentioned short note regarding 20CR.

Data and area
Acknowledging the geological context of a site-specific analysis, I would suggest a classical structure 2.1 Area/Leba Dunes, 2.2 Climatological characteristics, 2.3 Meteo data, 2.4 Dunes

We changed the order accordingly.

2.1. Meteorological data
I think it is enough to write "we use a numerically downscaled reanalysis dataset … "
with two or three sentences which reanalysis and regional model, spectral nudging and
resolution used in coastdat2. For details you can refer to Geyer 2014. More relevant for
this study are the properties and validation of simulated wind (Weidemann 2014).

We shortened the explanation about coastDat2.

Page 3: line 29: "usually kept to a minimum". This is not correct, all available data is
used for each time step. Reanalysis hence gives the best possible estimate for each
time step which may lead to artifacts if the type, quality or number of observations
changes over time. Rather a frozen data assimilation scheme is used to minimize
these effects. For your region and time period, all the issues are not really relevant
after ~1980.

This sentence was deleted to shorten the description of coastDat2 addressing
your preceding comment.

Page 4: line 12: The imperfect NCEP forcing and its coarse resolution is another
relevant source of error here

We added: "Other errors may occur due to the imperfect forcing data set NCEP
and its coarse spatial resolution. "

Page 4, line 33: Could you give the typical size here (dominant sand fraction, µ m or a
range)?

We added: "The sands are fine-grained (with a diameter of 0.2 to 0.3 mm
(Ludwig, 2017)) and well-sorted…"

Page 5, line 16: What about wind? Reference here to Ludwig et al. for more detailed
seasonal information (Fig. 5 in that paper)

We do not want to rely solely on the weather information given by Ludwig et al.
(2017) as they  based their conclusions on only one station, where wind from
some directions is potentially perturbed by the nearby forest.
Therefore we added: "Regarding wind direction, the Baltic Sea area shows a
predominance of westerly and southwesterly winds for all seasons with a second
maximum for north-easterly winds during spring for mean and for extreme wind
speeds (Bierstedt, 2015). Similar wind climatology was obtained by Ludwig
(2017) and Ludwig et al. (2017) with observational data of one station located
close to the dunes."

Page 5: line 28: Is there not any experimental / theoretical range giving a rule of thumb
which wind speed can mobilize which grain size or mass? If so, you could use it to
physically verify the realism of your statistically estimated thresholds.

We added: "Ludwig et al. (2017) estimated this threshold to be 4.4 m/s for the

finest dry sands and 10 m/s for moist material. "

Page 5, line 31: compiled

   We changed this accordingly.

Statistical Methods
It is convincing and very nicely shown, that one barcode interval reflects one year in Ludwig et al. However, the separation into junks of three months for wind is in the end subjective and artificial wrt. to the annual dune activity. Consider e.g. that your seasonal correlation analysis may suffer from intra-seasonal changes in wind activity over time (e.g. Lehmann et al. 2011). I would hence suggest to add first a correlation analysis for the whole wind year (e.g. ONDJFM). You could then replace the JJA figures in the multi plot figures 4 and 5 with the full wind season. Then you can continue and show also whether the full wind season or rather a fraction of the season yields the highest correlation.

   We also re-did the analysis for the whole wind year (SONDJFM). However, the
   results in this case show lower correlations and less differentiated results for the
   black and white layer. Because one goal was to analyze seasonal differences, the
    seasonal  analysis was retained, although the results for the   whole wind year
   are now  discussed in the manuscript.

The same applies for the wind directions. The correlations for the wind octants in combination with the small sample size in this study may be quite sensitive to small random changes to the neighboring octant (as can be anticipated from the wind rose figures in Ludwig et al.). I think you should test in addition quadrants of 90◦(e.g. W= 225-315◦or SW=180-270◦). It is certainly interesting to make the detailed tests in this study but one aim should be to find the optimal setting with the best fit to the dune data rather than limiting it to very strict seasons and directions.

   The coarser separation into wind directions was, in a similar way, already
   prescribed by Ludwig et al. (2017), although they used a different wind data set
   (observations). The suggestion by the reviewer is indeed very logical, but
   unfortunately we face the limitation of the short record. If the wind data are
   further stratified according to finer directional bin resolution, the sample size for
   each direction bin will become really small, compromising the statistical analysis.

Page 7: line 2: Please explain in more detail how such a ratio or difference might look like and why, I cannot follow here.

   We have expanded this explanation in the new version according to following
   scheme:
   Old version: "In addition, we try to find an optimal ratio between the number of
   westerly and easterly winds that better describe the thickness of the black
   interval."

   New version: "Due to the described winnowing effect of easterly winds (see Sect.
   2.1.2), we additionally investigated the idea of an optimal ratio between the
   number of westerly and easterly winds which promotes the thickness of black
   layers. "

What are the x-axis units in Fig. 8?

> Old figure caption: "Scatter plot of the difference between westerly (W, SW, NW) and easterly (E, SE, NE) winds and the black interval thickness. The red line shows..."

> New figure caption: "Scatter plot of the difference between the number of westerly (W, SW, NW) and easterly (E, SE, NE) winds and the black interval thickness. The red line shows..."

Page 7, line 3: I see the point of exploring the outcome of a LOESS fit here. But the result does not look useful. Based on Fig. 8, a linear regression (digitizing your data in Fig. 8, I got y=0.00053x+3.388; r 2=34%; p=0.0012) looks more convincing although it remains unclear to me, what it means. With the low sample size, LOESS regression is very sensitive to outliers. As it does not yield any equation, the fit cannot be reproduced by others without having the original data. I would therefore suggest to stick to a linear fit, give confidence intervals and explain the outcome.

> Maybe the reason for the application of LOESS was not clear enough in the old version of the manuscript. Our goal was to see if there is an optimal relation between westerly and easterly winds which promotes the thickness of black layers, so that smaller or larger ratios would produce also a smaller thickness. For this, we need to identify a nonlinear link between this ratio and the layer thickness. Unfortunately the obtained results are not robust so that such an optimal relation cannot be confirmed. We try to explain this more clearly in the new version. In contrast, a linear regression would not serve our purposes in this case. This is related to the previous comment of the reviewer. A linear regression would only show the influence of the difference between westerly and easterly winds. We already know that westerly winds are the driving force as the dune is moving towards the east. Hence, this would not give us new information.

Page 8, line 3: "slight positive correlation" – Which value? Give a p-value.

> Old version: "The colder seasons winter (DJF) and spring (MAM) show slight positive correlations for both intervals,..."

> New version: "The colder seasons winter (DJF) and spring (MAM) show slight, albeit not significant at the 95% level, positive correlations for both layers (DJF; r=0.17-0.23 and MAM; r=0.19-0.24),..."

Page 8, line 4: "indicates an increasing bar thickness during wetter periods". And how does that match with soil wetness and compactness mentioned before? I think this only tells you that more storms co-vary with more rain, but there is no causation more rain = more sand transport. This should be at least discussed if the low sample size does not allow a comparison like drier storm seasons vs. wetter storms seasons in comparison to the barcode. Maybe you could make a quick test for your period if/how wind above your chosen threshold is correlated with precipitation and temperature.

> We do not totally agree with the reviewer on this point, since conditional on the small sample size we have indeed attempted to separate the effect of

temperature and precipitation from the influence of wind on dune mobilization. The correlation between the layers thickness and the number of days with specific wind directions has been estimated including or excluding frost days or rainy days. As explained in the text, there are some differences in the expected direction, i.g. excluding frost days includes the correlation to the wind. This aspect is already discussed in our 'Discussion and Conclusion' section on page 11 line 24 -29. The reviewer is correct that using only statistical analysis it is not possible to totally disentangle the effect of wind and of rain if both co-vary, and when the differences in correlations are not very big, but In our discussion we do rely on field measurements and physical insights.

We added another sentence (underlined) and hope to better address this comment:
"Regarding precipitation, the results showed positive signs for the white and black bars for winter (DJF) and spring (MAM). Boŕowka (1980) stated that some rain might improve the transport due to turbulence, which makes more sand grains available.
We argue that the influence of precipitation on sand transport, and hence on the dune processes, depends on the seasonal wind conditions. For example it might be possible that precipitation and wind co-vary, which is especially likely during winter and spring when stronger cyclones come into the Baltic Sea region. Ludwig et al. (2017) describe a secondary dune on top of the primary dune consisting of the white and black interval. These secondary dunes seem to be affected by precipitation due to erosion. This idea is supported by our results and shows that in wetter seasons the secondary dunes might be eroded into the primary dune and hence results in thicker dune intervals."

Page 8, line 5: "non-negligible" – Please give a value and p-value here. Why only the black?

Old version: "Autumn is the only season showing some non-negligible correlation for black intervals (0.33)."

New version: "Autumn is the only season showing a non-negligible, albeit not significant correlation for black layers (r=0.33; p=0.09). "

Page 9, line 17: "this season and direction" – It makes sense to use the best combination but you should reconsider whether the best combination might not be the full wind season (e.g. SONDJFM) as mentioned before.

We calculated the results for the whole wind year (SONDJFM). However, the whole wind year results show lower correlation values and less differentiated results for the black and white layer. Because one goal was to analyze seasonal differences the seasonal analysis was retained and the results for the whole wind year are mentioned in the text.

As you mentioned the dunes in Lithuania, it would make sense to also provide your regression model for re-use or reproduction of results.

We added slope and intersect values in Table 1.

Page 10, line 2: To which extent could you use the deviation of the black-white ratio from being relatively equal (~1) to say sth. about years of more easterly or more westerly years? I did not really get that point from the manuscript.

> This part of the manuscript is about the question whether there exist an optimal ratio between westerly and easterly winds which might promote the thickness of the black layer. We wrote on page 7, line 5-7: "Due to the described winnowing effect of easterly winds (see Sect. 2.1.2), we additionally investigated the idea of an optimal ratio between the number of westerly and easterly winds which promotes the thickness of black layers. The idea is that smaller or larger ratios would produce thinner or thicker black layers."
> So this section is not about the black-white ratio, but on the possibility to record the ratio westerlies-to-easterlies on the thickness of the black layer.

Page 10, line 10-13: This fit makes little sense. Please replace LOESS with a linear fit and give the equation, $r_2$ and p-value (should be very close to what I wrote above). There is indeed no optimum (why should there?) but a linear fit is highly significant. What could that mean?

> Please, see answer to this comment above.

Page 10, line 17-20: The link between wind climate and sea-level is a bit more complicated depending on the region and timescale of wind/sea-level co-variations. The description here is too vague and some references should be given in addition. Note that most readers do not know anything about sea-level variations of the Baltic Sea.

> This section was indeed too short in the old version and we expanded it to make it clearer also for readers not familiar to sea-level variability. Originally, it was intended to indirectly support the link between the dune layers and wind, in view of the dearth of wind observations in this area.

Page 10, line 29: And how does the link of the wind forcing look like= How can it be explained?

> We hope that the previous addition about the link between wind and Baltic Sea level already clarifies this question .

Page 11, line 15-16: Rather than being suspicious about coastdat2 here, I would highlight that the positive link to sea-level is very useful as sea-level data goes further back in time than reanalysis and might be also more reliable than spurious trends in 20CR (Krueger et al. 2013), which do not affect yet the short period in this study.

> We agree with the reviewer and added information about the longer time span of sea-level data. Nevertheless in our case it was not the intention to use these data, because the dunes only span 26 years. We wanted to make our results more robust by also shiwung the link of the dune layers to other observational data in addition to the reanalysis. Regarding the 20CR reanalysis, we think another note regarding 20CR might be confusing.

Page 11, line 25-26: Very speculative. With "some rain" it might be true but not with

more rain. If the "some rain" effect would be important, you should expect to get a negative correlation in your evaluation, but it is positive. I would add that more rain might just co-vary with more windy conditions. The mentioned erosion is also a very good point here.

> We were here quoting a previous study by  Borówka, so that this reasoning is not entirely  based on our speculation. We have rephrased the paragraph:
> Old version: "Borówka (1980) stated that some rain might improve the transport due to turbulence, which makes more sand grains available. We argue that the influence of precipitation on sand transport, and hence the dune processes, depends on the seasonal wind conditions."
>
> New version: "Borówka (1980) stated that some rain might improve the transport due to turbulence, which makes more sand grains available. We argue that the influence of precipitation on sand transport, and hence on the dune processes, depends on the seasonal wind conditions. For example it might be possible that precipitation and wind co-vary, which is especially likely during winter and spring when stronger cyclones come into the Baltic Sea region. "

Page 12, line 5: The p-values and adding an analysis of the full wind season ONDJFM might lead to an even more robust conclusion.

> For the sake of clarity we would rather prefer to mark significant (by the 0.05 significance level) results with a *. This was also a suggestion of the first reviewer.

Figures:
Fig. 3+4+5: Use consistent tick marks on the y-axis. What means "- mv" in the figure titles? For all bar charts, you could consider using white, black and grey for white, black and mixed intervals. This would make it more intuitive.

> We will change this accordingly.

Figure 8: I would rather use a linear fit. What are the units on both axis?

> Please see answers above.

Figure 9: If possible, use bigger symbols for the gauge locations.

> We enlarged the symbols.

Table 1: Why not give the regression model (slope, intersect) in addition, also p-values?

> We added the slope and intersect of our regression model. Regarding p-values, please, see above.

Additional references:

Christensen, O. B.; Kjellström, E. & Zorita, E.: Projected Change—Atmosphere. In: The BACC II Author Team (Eds.): Second Assessment of Climate Change for the Baltic Sea Basin, Springer International Publishing, 2015, 217-233, doi:10.1007/978-

3-319-16006-1_11

Lehmann, A.; Getzlaff, K. & Harlaß, J. (2011): Detailed assessment of climate vari-ability in the Baltic Sea area for the period 1958 to 2009. Clim. Res., 46, 185-196, doi:10.3354/cr00876

Lindenberg J, Mengelkamp H-T, Rosenhagen G (2012): Representativity of near sur-face wind measurements from coastal stations at the German Bight. Met Z 21:99-106.

Rutgersson, A., Jaagus, J., Schenk, F., Stendel, M., Bärring, L., Briede, A., Claremar, B., Hanssen-Bauer, I., Holopainen, J., Moberg, A., Nordli, O., Rimkus, E., and Wibig, J.: Recent Change – Atmosphere. In: The BACC II Author Team (Eds.): Second Assess-ment of Climate Change for the Baltic Sea Basin. Springer International Publishing, 2015, 69-97, doi:10.1007/978-3-319-16006-1_4
**A wind proxy based on migrating dunes at the Baltic Coast: statistical analysis of the link between wind conditions and sand movement.**

Svenja E. Bierstedt[1], Birgit Hünicke[1], Eduardo Zorita[1], and Juliane Ludwig[2]

[1]Institute for Coastal Research, Helmholtz-Zentrum Geesthacht, Geesthacht, Germany
[2]Institute of Geology, University Hamburg, Hamburg, Germany

*Correspondence to:* Svenja Bierstedt (svenja.bierstedt@hzg.de)

**Abstract.** We statistically analyse the relationship between the structure of migrating dunes in the Southern Baltic and the driving wind conditions over the past 26 years, with the long-term aim of using migrating dunes as proxy for past wind conditions at interannual resolution.

The present analysis is based on the dune record derived from geo-radar measurements by (Ludwig et al., 2017). The dune system is located at the Baltic Sea coast of Poland and is migrating from west to east along the coast.  The dunes present layers with different thickness that can be absoluted dates at interannual timescales and whose thickness can be put in relation to seasonal wind conditions. To statistically analyse this record and calibrate it as a wind proxy  we used a gridded regional meteorological reanalysis data set (coastDat2)  covering the recent decades. Furthermore, the identified link between the dune annual layers and wind conditions was additionally supported by the co-variability between dune layers and observed sea-level variations in the Southern Baltic Sea.

We include precipitation and temperature into our analysis, in addition to wind, to learn more about the dependency between these three atmospheric factors and their common influence on the dune system. We set up a statistical linear model based on the correlation between the  frequency of days with  specific wind conditions in a given season and dune migration velocities derived for that season. To some extent, the dune  records can be seen analogous to tree ring width record, and hence we used a proxy-validation method usually applied in dendrochronology, namely the cross-validation with the leave-one-out-method, when the observational record is short. The revealed correlations between the wind record from the reanalysis and the  wind record derived from the dune structure is in the range  between 0.28 and 0.63 ~~. Thus, our study verifies that dendrochronological methods can be applied to validate migrating dunes as a wind proxy, deriving acceptable validation values.
[revised manuscript text omitted]